# Baseline factors predicting the need for corneal crosslinking in patients with keratoconus

**Naoko Kato**[1,2]*, **Kazuno Negishi**[2], **Chikako Sakai**[2], **Kazuo Tsubota**[2,3]

**1** Minamiaoyama Eye Clinic, Tokyo, Japan, **2** Department of Ophthalmology, School of Medicine, Keio University, Tokyo, Japan, **3** Tsubota Laboratory, Inc., Tokyo, Japan

* naokato@bc.iij4u.or.jp

## Abstract

**Data Availability Statement:** All relevant data are within the paper and its Supporting Information file.

**Funding:** This work was partly supported by funding from EyeLens Pte. Ltd., a distributor of the

### Introduction

The primary purpose of crosslinking is to halt the progression of ectasia. We retrospectively assessed the condition of keratoconus patients who were followed-up at least twice after the initial examination to evaluate keratoconus progression, to identify definitive factors to predict a later need for corneal crosslinking (CXL).

### Methods

The medical charts of 158 eyes of 158 keratoconus patients (112 males and 46 females; mean age, 27.8 ± 11.7 years), who were followed up at the Department of Ophthalmology, Keio University School of Medicine at least twice after the initial examination to evaluate keratoconus progression were retrospectively reviewed. Best-spectacle corrected visual acuity, intraocular pressure, steepest corneal axis on the anterior float (Ks), thinnest corneal thickness according to Pentacam® HR, and corneal endothelial cell density were assessed. Gender, age, onset age of keratoconus, history of atopic dermatitis, and Pentacam® indices were also recorded. CXL was performed when the eye showed significant keratoconus progression, an increase in the steepest keratometric value, or an increase in the spherical equivalent or cylinder power of the manifest refraction by more than 1.0 D versus the respective values 2 years prior. Predictor variables and the requirement for CXL were analyzed using logistic regression.

### Results

Fifty-eight eyes required CXL treatment. The best predictor of the requirement for CXL was patient age, followed by the Pentacam® Rmin (the minimum sagittal curvature evaluated by Pentacam®) value. The incidence of CXL was 86.4% in the < 20 years age group, with an Rmin of ≤ 5.73 mm, whereas 10.8% in the ≥ 27 years age group with an Rmin > 5.73 mm underwent treatment.

products and/or procedures of corneal crosslinking. The funders had no role in study design, data collection and analysis, decision to publish, or preparation of the manuscript. Outside the submitted work, Kazuo Tsubota reports his position as CEO of Tsubota Laboratory, Inc., Tokyo, Japan, a company producing a keratoconus treatment-related device. Tsubota Laboratory, Inc. provided support in the form of salary for KT, but did not have any additional role in the study design, data collection and analysis, decision to publish, or preparation of the manuscript.

**Competing interests:** Competing interests: Outside the submitted work, Kazuo Tsubota reports his position as CEO of Tsubota Laboratory, Inc., Tokyo, Japan, a company producing a keratoconus treatment-related device. Tsubota Laboratory, Inc. provided support in the form of salary for KT, but did not have any additional role in the study design, data collection and analysis, decision to publish, or preparation of the manuscript. This does not alter our adherence to PLOS ONE policies on sharing data and materials.

## Conclusions

An age of < 20 years and an Rmin value of $\leq 5.73$ mm predicted keratoconus progression and the requirement for CXL treatment in the near future.

## Introduction

Up until the end of the 20th century, keratoconus had been an incurable disease. The progression of the disease could not be halted, with corneal transplant being the only way to treat impaired visual function, and only in the most severe cases. However, the development of the corneal crosslinking (CXL) procedure by Wollensak et al. in 2003 [1] provided the means to arrest disease progression, such that keratoconus can be diagnosed and cured at an early stage.

The primary purpose of crosslinking is to halt the progression of ectasia. The best candidate for this therapy is an individual with keratoconus or post-refractive surgery ectasia who has documented disease progression. However, there are no definitive criteria for predicting keratoconus progression at present. The parameters that must be considered are changes in refraction (including astigmatism), uncorrected visual acuity, best spectacle-corrected visual acuity (BSCVA), and corneal shape and thickness (according to corneal topography or tomography) [2–6].

Widely accepted indications for CXL include an increase of 1.00 D or more in the steepest keratometry measurement, an increase of 1.00 D or more in the manifest cylinder, and an increase of 0.50 D or more in the manifest refraction spherical equivalent (MRSE) [7]. It takes several months to determine whether a patient meets the clinical criteria for CXL. Recently, Wisse et al. proposed a novel and easy-to-use CXL scoring system, the Dutch Crosslinking for Keratoconus (DUCK) score, for keratoconus patients; however, the DUCK score also requires maximum keratometry differences, with months between measurement intervals [8]. The disease may progress rapidly during the follow-up period, even while awaiting CXL [9]. Therefore, a method for determining the need for CXL in keratoconus cases on first examination is urgently needed.

In the present investigation, we retrospectively assessed the condition of keratoconus patients seen at our institute who were followed-up at least twice after the initial examination to evaluate keratoconus progression, in an attempt to determine definitive predictors of the future need for CXL treatment.

## Methods

This retrospective study followed the ethical standards of the Declaration of Helsinki and the study protocol was approved by the Institutional Review Board of the Keio University School of Medicine.

### Patients

The medical charts of 158 eyes of 158 keratoconus patients (112 males and 46 females; mean age, 27.8 ± 11.7 years), who visited the Department of Ophthalmology, Keio University School of Medicine from January, 2009 to August, 2018 at least twice, were investigated retrospectively. If the both eyes were affected, the right eye was investigated. The period between the initial and final visits varied from 6 weeks to 8.6 years (mean period, 2.61 ± 2.09 years). The study

protocol was approved by the Institutional Review Board of Keio University School of Medicine. Keratoconus was diagnosed based on corneal tomography, i.e., ectasia screening using the CASIA® device (Tomey, Aichi, Japan), and/or topographic keratoconus classification (TKC) using the Pentacam® HR instrument (Oculus, Wetzlar, Germany). Eyes with pellucid marginal degeneration, keratectasia after laser refractive corneal surgery, previous acute hydrops, or other diseases were excluded.

### Examinations

Examinations performed at the initial visit included a standard ophthalmic examination with measurement of the BSCVA, intraocular pressure (IOP), steeper corneal axis on the anterior float (Ks), baseline thinnest corneal thickness according to corneal tomography (Pentacam® HR), and corneal endothelial cell density (EM-3000; Tomey, Nagoya, Japan). Gender, age, onset age of keratoconus, and history of atopic dermatitis were also recorded.

### Indications for CXL

CXL treatment was applied to eyes with recently active keratoconus that showed significant keratoconus progression, i.e., an increase in the steepest keratometric value, spherical equivalent, or cylinder power of the manifest refraction of more than 1.0 D versus the equivalent value 2 years previously.

### Statistical analysis

Stepwise regression was carried out using JMP12 software (SAS Institute Inc., Cary, NC, USA). Independent associations between predictor variables and the requirement for CXL were analyzed using multiple logistic regression analysis. A p-value $< 0.05$ was considered to be statistically significant.

### Results

In total, 58 eyes required CXL and 100 eyes were followed-up without application of CXL. The multiple coefficient of determination ($R^2$) for estimating the probability of CXL was 0.208. The factor showing the strongest association with the requirement for CXL was age, which is well-known to be associated with keratoconus progression, followed by the Pentacam® Rmin measurement (Table 1). A significantly high close relationship between the probability of CXL and multiple linear regression value was demonstrated ($R^2 = 0.208$, $p < 0.001$), and the correspondent area under the receiver-operating characteristic (ROC) curve for probability of CXL was estimated to be 0.878. Regression graphs for these two factors are shown in Fig 1; significant correlations with the requirement for CXL are evident.

We then calculated the incidence of CXL in groups classified according to age and Rmin value. The cut-off value determined by the receiver operating characteristic (ROC) curves for the requirement for CXL were 27.0 years of age, and Rmin values of 5.73 mm, respectively. In total, 75.0% of eyes underwent CXL in the group aged $< 27$ years with an Rmin $\leq 5.73$ mm, while 10.8% of eyes underwent CXL in the group aged $\geq 27$ years with an Rmin $> 5.73$ mm.

The 50th percentiles for the requirement for CXL according to reverse estimation were 20.0 years of age. When we divided the younger group of patients, 86.4% of eyes underwent CXL in the group aged $< 20$ years with an Rmin $\leq 5.73$ mm, while 63.6% of eyes underwent CXL in the group aged 20–26 years with an Rmin $> 5.73$ (Table 2; Fig 2).

**Table 1. Relationship between CXL and baseline examination data (multiple logistic regression analysis).**

| Factors | Wald score | P-value | 95%CI |
|---|---|---|---|
| Age | 11.695 | 0.001 | 0.080–0.446 |
| Gender | 0. 037 | 0. 847 | -0.510–0.919 |
| History of atopic dermatitis | 2.705 | 0.100 | -0.832–0.994 |
| Age on diagnosis | 6.275 | 0.012 | -0.305–0.071 |
| BSCVA | 0.482 | 0.488 | -2.936–2.898 |
| Manifest cylinder value | 1.544 | 0.214 | -0.265–0.389 |
| Manifest spherical equivalent | 0.848 | 0.357 | -0.237–0.152 |
| IOP | 0.044 | 0. 834 | -0.218–0.204 |
| K2 on the anterior float | 2.001 | 0.157 | -2.427–1.384 |
| K2 on the posterior float | 1.678 | 0.195 | -1.787–2.956 |
| Total K2 | 2.112 | 0.146 | -1.430–2.362 |
| ISV | 5.802 | 0.016 | -0.112–0.142 |
| IVA | 7.036 | 0.008 | -12.199–2.302 |
| KI | 4.665 | 0.031 | -2.816–39.619 |
| CKI | 2.936 | 0.087 | -7.378–35.953 |
| IHA | 0.229 | 0.632 | -0.013–0.032 |
| IHD | 6.455 | 0.011 | -29.139–14.853 |
| Rmin | 7.380 | 0.007 | -1.014–6.718 |
| TKC; 2 and below vs 2–3 or more | 7.828 | 0.005 | -57498.416–57456.610 |
| TKC; 0 vs possible or more | 8.607 | 0.014 | -2.650–4.141 |
| TKC; 1–2 or below vs 2 | 8.996 | 0.029 | -1.244–5.877 |
| TKC; possible vs 1 and 1–2 | 9.389 | 0.052 | -5.458–2.029 |
| TKC; 1 vs 1–2 | 9.500 | 0.091 | -3.733–1.680 |
| TKC; 2–3 vs 3 or more | 8.350 | 0.015 | -1.897–3.441 |
| TKC; 3 vs 3–4 and 4 | 8.439 | 0.038 | -1.691–5.393 |
| TKC; 3–4 vs 4 | 8.439 | 0.077 | -10.601–2.494 |
| CCT | 0.952 | 0.329 | -0.037–0.028 |
| TCT | 1.879 | 0.171 | -0.037–0.026 |

CXL, corneal cross-linking; BSCVA, best spectacle-corrected visual acuity; IOP, intraocular pressure; K2, the steepest keratometric value indicated by Pentacam® HR; ISV, index of surface variance; IVA, index of vertical asymmetry; KI, keratoconus index; CKI, center KI; IHA, index of height asymmetry; IHD, index of height decentration; Rmin, the minimum sagittal curvature evaluated by Pentacam®; TKC, topographic keratoconus classification; CCT, central corneal thickness; TCT, thinnest corneal thickness; 95% CI, 95% confidence interval.

## Discussion

The present investigation showed that a young age and small Rmin value at the initial examination predicted the requirement for CXL in the future. Patients aged < 20 years with Rmin ≤ 5.73 mm had a significantly higher probability of needing CXL treatment in the future. On the other hand, patients aged > 27 years with Rmin > 5.73 mm had no requirement for CXL later in life.

Keratoconus is well known to progress in younger patients. Early onset keratoconus tends to progress at a faster rate, and is more likely to require corneal transplant, compared with late onset cases [10,11]. Therefore, many clinicians recommend that CXL be performed immediately in younger keratoconus patients [12–14], which accords with our finding of an association of younger age with the requirement for CXL.

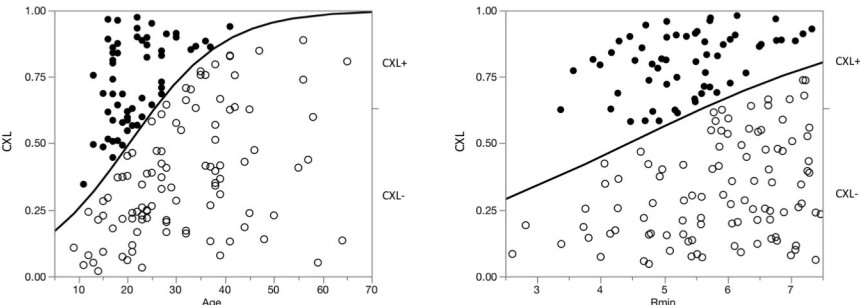

**Fig 1. Relationship of the requirement for corneal crosslinking (CXL) with age and Rmin at the first visit in multiple logistic regression analysis (left, age; right Rmin).** Left, the relationship between age and requirement for CXL was obtained by logistic multiple regression analysis of data for 58 subjects who underwent CXL (solid circles) and 100 subjects who did not (open circles). The abscissa and ordinate are the logarithmic values of age and CXL probability, respectively. Right, the relationship between Rmin and requirement for CXL obtained through logistic multiple regression analysis. The abscissa and ordinate are logarithmic values of Rmin (mm) and CXL probability, respectively.

The second predictor of the requirement for CXL in this study was Rmin, i.e., the minimum sagittal curvature as evaluated by Pentacam®, which corresponds to the maximum kerato-metric value (Kmax; 5.73 mm is equivalent to 58.9 D). Patients aged < 20 years having a Rmin value ≤ 5.73 mm should be treated with CXL immediately.

Ferdi et al [15] conducted a systemic review and meta-analysis on keratoconus progression and concluded that patients aged less than 17 years, and those with a Kmax > 55 D, are at significantly greater risk of keratoconus progression. When we classified eyes according to an Rmin cut-off value of 6.13 mm (equivalent to 55.0 D) and age cut-off of 17 years, according to Ferdi's study, the incidence of CXL was 85.7% in the eyes of the younger age group with the lower Rmin values. Thus, based on these results, we propose that young patients aged < 20 years with more than moderate keratoconus (at least with a Kmax ≥ 58.9 D) should undergo CXL immediately after diagnosis.

In contrast, patients aged ≥ 27 years had a lower requirement for future CXL. In particular, only 8.5% of eyes with Rmin > 5.73 mm (equivalent to 58.9 D) required CXL. Moreover, there were no eyes with Rmin > 6.53 mm (equivalent to 51.7 D) required CXL in this age group (Fig 2). Some researchers have suggested that keratoconus may progress even beyond 30 years of age [16–18]; however, they did not categorize patients with respect to the Rmin value at baseline. The present results indicate that mild to moderate keratoconus at the age of 27 years or older should be followed-up; however, frequent follow-up visits are not necessary. Eyes with severe keratoconus (especially with a Rmin ≤ 6.73 mm [58.9 D]) should be followed-up closely, even beyond the age of 27 years, as these patients may also develop acute hydrops [17].

**Table 2. Incidence of CXL according to age and Rmin at the first examination.**

|  | 26 years or younger | | 27 years or older |
|---|---|---|---|
|  | 19 years or younger | 20–26 years-old |  |
| Rmin < 5.73 mm | 75.0% | | 25.0% |
|  | 86.4% | 63.6% | |
| Rmin > 5.73 mm | 29.3% | | 10.8% |
|  | 38.1% | 20.0% | |

Rmin, the minimum sagittal curvature evaluated by Pentacam®.

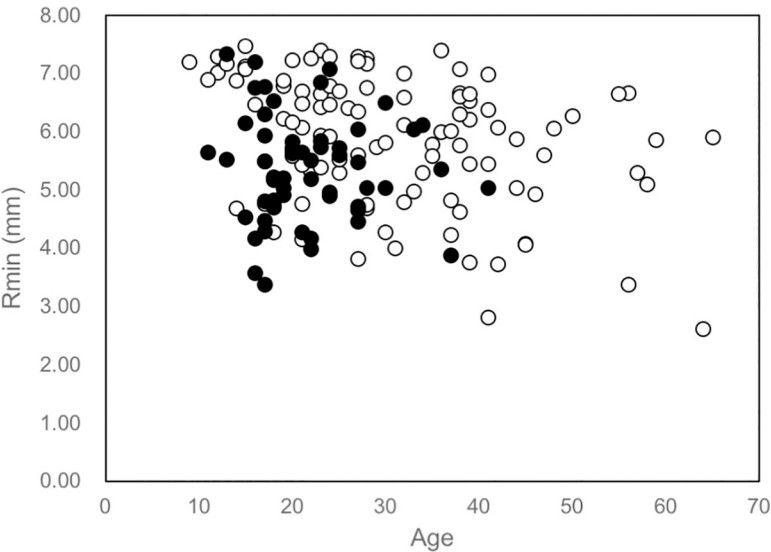

**Fig 2. Scatter diagram of the relationship between age and Rmin at the initial visit and the requirement for CXL thereafter.** A younger age and smaller Rmin were associated with the need for CXL treatment. Solid circles indicate the 58 subjects who underwent CXL treatment; open circles indicate the 100 subjects who received no CXL treatment.

Other factors are suspected to influence keratoconus progression; however, here, gender, history of atopic dermatitis, a steeper corneal axis on the anterior and posterior float, IOP, and corneal thickness were not correlated significantly with the future requirement for CXL. We did not analyze the corneal biomechanical response [19–22], as the instruments required for this evaluation were not commercially available when our examinations began. We expect that corneal biomechanical response evaluation will facilitate early diagnosis of progressive keratoconus and prediction at the initial visit of the future requirement for CXL.

The limitation of the present retrospective study was that the follow-up period of the enrolled cases varied from 6 months to 8.6 years. If we could investigate only the cases that were followed for 2 years or more, we should have obtained the more precise probability for requirement for CXL. However, we recommend the CXL as soon as possible, when we suspect the progression of keratoconus especially for young patients.

At present, keratoconus progression can be evaluated only by repeated examination of visual acuity and corneal topographic/tomographic changes over time. However, the present investigation proposes two new predictors of the progression of keratoconus and need for CXL treatment in the near future: an age of less than 20 years and an Rmin value of $\leq 5.73$ mm (Kmax $\geq 58.9$ D).

## Supporting information

**S1 Data.**
(PDF)

## Acknowledgments

The authors wish to thank Satoru Inoda, M.D. at Department of Ophthalmology, Jichi Medical University for his valuable suggestion about statistical analysis, and also thank Hidemasa Torii, M.D., Ph.D. and Ms. Sachiko Masui at Department of Ophthalmology, Keio University School of Medicine for their valuable discussion and help for data collection and storage.

## Author Contributions

**Conceptualization:** Naoko Kato.

**Data curation:** Naoko Kato, Chikako Sakai.

**Formal analysis:** Naoko Kato, Chikako Sakai.

**Funding acquisition:** Kazuno Negishi.

**Investigation:** Naoko Kato, Chikako Sakai.

**Methodology:** Naoko Kato.

**Project administration:** Naoko Kato, Kazuno Negishi, Kazuo Tsubota.

**Resources:** Naoko Kato, Kazuno Negishi.

**Supervision:** Kazuno Negishi, Kazuo Tsubota.

**Validation:** Kazuno Negishi, Chikako Sakai.

**Visualization:** Naoko Kato.

**Writing – original draft:** Naoko Kato.

**Writing – review & editing:** Kazuno Negishi, Chikako Sakai, Kazuo Tsubota.

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
