## [Decision Letter · Decision Letter 0]

8 Jan 2020

PONE-D-19-25186

Baseline factors predicting the need for corneal crosslinking in patients with keratoconus

PLOS ONE

Dear Dr Kato,

Thank you for submitting your manuscript to PLOS ONE. After careful consideration, we feel that it has merit but does not fully meet PLOS ONE’s publication criteria as it currently stands. Therefore, we invite you to submit a revised version of the manuscript that addresses the points raised during the review process.

We would appreciate receiving your revised manuscript by Feb 16 2020 11:59PM. To enhance the reproducibility of your results, we recommend that if applicable you deposit your laboratory protocols in protocols.io, where a protocol can be assigned its own identifier (DOI) such that it can be cited independently in the future. For instructions see: http://journals.plos.org/plosone/s/submission-guidelines#loc-laboratory-protocols

We look forward to receiving your revised manuscript.

Kind regards,

Yu-Chi Liu, M.D

Academic Editor

PLOS ONE

Journal Requirements:

'NO'

Please provide an amended Funding Statement that declares *all* the funding or sources of support received during this specific study (whether external or internal to your organization) as detailed online in our guide for authors at http://journals.plos.org/plosone/s/submit-nowPlease state what role the funders took in the study.  If any authors received a salary from any of your funders, please state which authors and which funder. If the funders had no role, please state: "The funders had no role in study design, data collection and analysis, decision to publish, or preparation of the manuscript."

'NO'

a. Please state any Competing Interests. If you have no competing interests, please state "The authors have declared that no competing interests exist.", as detailed online in our guide for authors at http://journals.plos.org/plosone/s/submit-now

Reviewers' comments:

Reviewer's Responses to Questions

**Comments to the Author**

1. Is the manuscript technically sound, and do the data support the conclusions?

Reviewer #1: Yes

Reviewer #2: Yes

Reviewer #3: Partly

2. Has the statistical analysis been performed appropriately and rigorously? 

Reviewer #1: Yes

Reviewer #2: Yes

Reviewer #3: N/A

3. Have the authors made all data underlying the findings in their manuscript fully available?

Reviewer #1: Yes

Reviewer #2: Yes

Reviewer #3: Yes

4. Is the manuscript presented in an intelligible fashion and written in standard English?

Reviewer #1: Yes

Reviewer #2: Yes

Reviewer #3: No

5. Review Comments to the Author

Reviewer #1: 1. In the Abstract, authors should explain the meaning of Rmin value.

2. In the Abstract, authors should add the method of logistic regression analysis.

3. When using both eyes for the subject, there is a bias, so one eye should be used.

4. How do you determine the presence of CXL if follow-up visit periods are different? If the follow-up periods are different, it is difficult to examine the number of CXL for the patients.

5. Write the equipment name, country etc. for the calculating corneal endothelial cell density.

6. Authors should explain how to conduct univariate- and multivariate logistic regression analysis. It seems that authors only conducted fully adjusted logistic regression analysis. Authors should confirm the regression multicollinearity and decide the risk factor for the analysis.

7. Authors should add 95% CI in the Table 1.

8. Adjust the number of digit of P value in the Table1.

9. Authors should explain how to examine the cut-off value for the age and Rmin. Do you use receiver operating characteristic (ROC) curves?

10. Unify the description of 17, 18, 19 years old through the manuscript.

Reviewer #2: This is a very interesting piece of work with potentially immediate clinical application. My main concern is that the criteria for the definition of progression of KC may hugely confound the results - i.e the identified predictors of progression.

Abstract:

' CXL was performed when the eye showed significant keratoconus progression, an increase in the corneal axis, or an increase in the spherical equivalent or cylinder power of the manifest refraction by more than 1.0 D versus the respective values 2 years prior.'

Please define: significant keratoconus progression, increase in corneal axis.

In the main text it is stated that: 'The period between the initial and final visits varied from 2 weeks to 8.6

years (mean period, 2.60 ± 2.09 years)'

This contradicts the above statement in the abstract. In addition, I have reservations whether progression actually develops and can be detected within a 2 week period. Any change within such a short period most likely represents variation and repeatability errors in repeat corneal topography scans or refraction.

Identifying progression and thus the group that had cross-linking is key to the validity of the study. It is worth repeating the analysis with more well defined criteria of progression. A start could be to exclude cases that had a follow-up interval of less than a few months (? 3 or 4 months) in order to exclude cases that may have not had true progression.

Reviewer #3: The manuscript is important to a clinical audience but is beyond the scope of the general PlOSOne reader. There are only 2 figures and the data is only accessible to researchers who meet the criteria for access to confidential data and that limitation is not described. There are numerous issues of incorrect tense and PloSONE does not copyedit manuscripts.

6. PLOS authors have the option to publish the peer review history of their article (what does this mean?). If published, this will include your full peer review and any attached files.

Reviewer #1: No

Reviewer #2: No

Reviewer #3: No

---

## [Author Response · Author response to Decision Letter 0]

19 Feb 2020

Reviewer #1: 1. In the Abstract, authors should explain the meaning of Rmin value.

We added the explanation of Rmin in the abstract (page 2, line 36-37). 

2. In the Abstract, authors should add the method of logistic regression analysis.

We added the sentence, “Predictor variables and the requirement for CXL were analyzed using logistic regression.” to the method in the abstract (page 2, line 33-34). 

3. When using both eyes for the subject, there is a bias, so one eye should be used.

We appreciate your advice. We recalculate with the data using only one eye for the subject. We basically analyzed the right eye, when the subject diagnosed as keratoconus on both eyes, and if the subject had keratoconus only on one eye, we used the affected eye. In addition, we also removed one case that was followed twice but with only 2 weeks of the interval. According to these changes, we rewrote the details of data and p-value throughout the manuscript, even though the conclusion was not changed. 

4. How do you determine the presence of CXL if follow-up visit periods are different? If the follow-up periods are different, it is difficult to examine the number of CXL for the patients.

We have decided to perform CXL based on the significant progression of keratoconus, i.e., an increase in the steepest keratometric value, spherical equivalent, or cylinder power of the manifest refraction of more than 1.0 D versus the equivalent value 2 years previously. If the patients have rapid progression between the short period, e.g. a few months, we decided to perform CXL. While the present study was retrospective, we could not set the observation time for all individuals, however, all patients revealed more than 1 of the above conditions. 

5. Write the equipment name, country etc. for the calculating corneal endothelial cell density.

We added the required information about the measurement of corneal endothelial cell density, “EM-3000; Tomey, Nagoya, Japan” on page 6, line 92. 

6. Authors should explain how to conduct univariate- and multivariate logistic regression analysis. It seems that authors only conducted fully adjusted logistic regression analysis. Authors should confirm the regression multicollinearity and decide the risk factor for the analysis.

We appreciate the reviewer’s comment. We added the outcomes of the regression multicollinearity to decide the risk factors (page 7, line 110 – 113). 

7. Authors should add 95% CI in the Table 1.

We added 95% CI in the table 1. 

8. Adjust the number of digit of P value in the Table1.

We appreciate your advice. I adjusted the number of digit of p value in the Table 1. 

9. Authors should explain how to examine the cut-off value for the age and Rmin. Do you use receiver operating characteristic (ROC) curves?

According to the above-mentioned changes of the number of objects, we reperform the statistical analysis, and found the cut-off value for the age and Rmin as 27 years old and 5.73 mm, respectively. Therefore, we rewrote the manuscript following these outcomes. 

10. Unify the description of 17, 18, 19 years old through the manuscript.

We appreciate your advice. We recalculated and found the cut-of value for the age as 20 years-old, so we unified the description to 20 years old. 

Reviewer #2: This is a very interesting piece of work with potentially immediate clinical application. My main concern is that the criteria for the definition of progression of KC may hugely confound the results - i.e, the identified predictors of progression.

Abstract:

' CXL was performed when the eye showed significant keratoconus progression, an increase in the corneal axis, or an increase in the spherical equivalent or cylinder power of the manifest refraction by more than 1.0 D versus the respective values 2 years prior.'

Please define: significant keratoconus progression, increase in corneal axis.

We thank you for your valuable advice. We corrected to the phrase to “an increase in the steepest keratometric value” (page 2, line 31). 

In the main text it is stated that: 'The period between the initial and final visits varied from 2 weeks to 8.6 years (mean period, 2.60 ± 2.09 years)'

This contradicts the above statement in the abstract. In addition, I have reservations whether progression actually develops and can be detected within a 2 week period. Any change within such a short period most likely represents variation and repeatability errors in repeat corneal topography scans or refraction.

Identifying progression and thus the group that had cross-linking is key to the validity of the study. It is worth repeating the analysis with more well defined criteria of progression. A start could be to exclude cases that had a follow-up interval of less than a few months (? 3 or 4 months) in order to exclude cases that may have not had true progression.

We thank you for your comment. We usually follow the patients every 2 or 3 months. Only 1 case has followed only at 2 weeks after the initial visit, so that we excluded this case and reanalyzed the statistics. Other cases were followed longer duration; from 6 weeks to 8.6 years (page 5, line 81-82).

Reviewer #3: The manuscript is important to a clinical audience but is beyond the scope of the general PlOSOne reader. There are only 2 figures and the data is only accessible to researchers who meet the criteria for access to confidential data and that limitation is not described. There are numerous issues of incorrect tense and PloSONE does not copyedit manuscripts.

We appreciate the reviewer and added the limitation of this study (page 12, line 194-198). We would like to emphasize that our present study is a retrospective study following more than 150 patients with keratoconus years-long with sets of ophthalmic examinations including precise corneal tomography. We believe that this must be important outcomes for many clinicians who examine and treat the patients with keratoconus.

Others:

We added the affiliation of the last author, Prof Kazuo Tsubota (Page 1, line 9 and 13).

---

## [Decision Letter · Decision Letter 1]

25 Mar 2020

Baseline Factors Predicting the Need for Corneal Crosslinking in Patients with Keratoconus

PONE-D-19-25186R1

Dear Dr. Kato,

We are pleased to inform you that your manuscript has been judged scientifically suitable for publication and will be formally accepted for publication once it complies with all outstanding technical requirements.

With kind regards,

Yu-Chi Liu, M.D

Academic Editor

PLOS ONE

Additional Editor Comments (optional):

Reviewers' comments:

Reviewer's Responses to Questions

**Comments to the Author**

1. If the authors have adequately addressed your comments raised in a previous round of review and you feel that this manuscript is now acceptable for publication, you may indicate that here to bypass the “Comments to the Author” section, enter your conflict of interest statement in the “Confidential to Editor” section, and submit your "Accept" recommendation.

Reviewer #1: All comments have been addressed

Reviewer #2: All comments have been addressed

Reviewer #3: All comments have been addressed

2. Is the manuscript technically sound, and do the data support the conclusions?

Reviewer #1: Yes

Reviewer #2: Yes

Reviewer #3: Yes

3. Has the statistical analysis been performed appropriately and rigorously? 

Reviewer #1: Yes

Reviewer #2: Yes

Reviewer #3: (No Response)

4. Have the authors made all data underlying the findings in their manuscript fully available?

Reviewer #1: Yes

Reviewer #2: Yes

Reviewer #3: Yes

5. Is the manuscript presented in an intelligible fashion and written in standard English?

Reviewer #1: Yes

Reviewer #2: Yes

Reviewer #3: Yes

6. Review Comments to the Author

Reviewer #1: This manuscript "Baseline Factors Predicting the Need for Corneal Crosslinking in Patients with Keratoconusis" is ready for the publish on PLOS ONE.

Reviewer #2: (No Response)

Reviewer #3: The authors have responded to the comments of the 3 reviewers in a careful response. In the future it would be easier if the authors marked the changes so that they can be cross-referenced with the comments.

The manuscript is a retrospective study but could benefit the population of readers.

7. PLOS authors have the option to publish the peer review history of their article (what does this mean?). If published, this will include your full peer review and any attached files.

Reviewer #1: No

Reviewer #2: No

Reviewer #3: No

---

## [Editor Report · Acceptance letter]

1 Apr 2020

PONE-D-19-25186R1 

Baseline Factors Predicting the Need for Corneal Crosslinking in Patients with Keratoconus 

Dear Dr. Kato:

I am pleased to inform you that your manuscript has been deemed suitable for publication in PLOS ONE. Congratulations! Your manuscript is now with our production department. 

With kind regards,

on behalf of

Dr. Yu-Chi Liu 

Academic Editor

PLOS ONE